# In Vivo Modification of Microporous Structure in Bacterial Cellulose by Exposing *Komagataeibacter xylinus* Culture to Physical and Chemical Stimuli

**DOI:** 10.3390/polym14204388

**Published:** 2022-10-18

**Authors:** Yolanda González-García, Juan C. Meza-Contreras, José A. Gutiérrez-Ortega, Ricardo Manríquez-González

**Affiliations:** 1Departamento de Madera, Celulosa y Papel, CUCEI, Universidad de Guadalajara, Km 15.5, Carretera Guadalajara-Nogales, Las Agujas, Zapopan 45020, Jalisco, Mexico; 2Departamento de Química, CUCEI, Universidad de Guadalajara, Blvd. Marcelino García Barragán # 1421, Esq. Calzada Olímpica, Guadalajara 44430, Jalisco, Mexico

**Keywords:** bacterial cellulose, magnetic field, UV light, chloramphenicol, salinity, porosity

## Abstract

Bacterial cellulose (BC) samples were obtained in a static culture of *K. xylinus* under the effect of a low-intensity magnetic field, UV light, NaCl, and chloramphenicol. The effect of such stimuli on the amount of BC produced and its production rate, specific area, pore volume, and pore diameter were evaluated. The polysaccharide production was enhanced 2.28-fold by exposing *K. xylinus* culture to UV light (366 nm) and 1.7-fold by adding chloramphenicol (0.25 mM) to the medium in comparison to BC control. All the stimuli triggered a decrease in the rate of BC biosynthesis. BC membranes were found to be mesoporous materials with an average pore diameter from 21.37 to 25.73 nm. BC produced under a magnetic field showed the lowest values of specific area and pore volume (2.55 m^2^ g^−1^ and 0.024 cm^3^ g^−1^), while the BC synthesized in the presence of NaCl showed the highest (15.72 m^2^ g^−1^ and 0.11 cm^3^ g^−1^). FTIR spectra of the BC samples also demonstrated changes related to structural order. The rehydration property in these BC samples is not mainly mediated by the crystallinity level or porosity. In summary, these results support that BC production, surface, and structural properties could be modified by manipulating the physical and chemical stimuli investigated.

## 1. Introduction

Cellulose is one of the most abundant biopolymers on earth [1]. Its biosynthesis takes place in plants and in certain bacteria, particularly, with *Komagataeibacter xylinus* (*Acetobacter xylinum*) is the most widely studied strain due to its high productivity [2]. Bacterial cellulose (BC) is a nanostructured polymer organized at multiple levels (protofibrils–ribbons–membranes) [3]; it is highly pure [4] and shows peculiar properties, including high mechanical strength (tensile strength and elasticity), a high degree of polymerization, high crystallinity, great water-holding capacity, high porosity and surface area, and excellent biological affinity and biodegradability. Those features propose it as a biomaterial suitable for multiple applications [5], such as diet food, gelling agents, functionalized paper sheets, a fire-retarding agent, medical pads, plastic composite supports, acoustic diaphragms, the textile industry, artificial blood vessels, and artificial skin [6]. Such a wide view of applications of BC brings big interest towards its production with defined structural features that can be controlled by manipulating the culture conditions of BC-producing strains. In this regard, the biosynthesis of BC (yield, productivity, and production rate) is mainly affected by the composition of the culture medium used, pH, temperature, dissolved oxygen, and the type of culture employed (static or agitated). The effect of those factors on BC properties, such as porosity, has also been studied. Pore size and surface area of native BC are important properties, not only for developing novel functional biomaterials for tissue engineering but also for immobilizing and storage of various compounds, such as proteins (enzymes) and drugs, for separation processes (membranes), sorption, and catalytic purposes [7,8,9]. Increasing the surface area of the native BC membrane is essential to improve its performance in applications such as sorption of contaminants and toxic compounds, as well as for immobilization of enzymes and delivery of drugs. This has been achieved by introducing foreign substances (aluminum, iron, silica gel, and glass beads, among others) or using porogen/particle-leaching techniques (salt, paraffin, ice, gelatin, and sugar, among others) into a BC network [10,11,12,13]. It has also been reported that BC with controlled porosities could be obtained by varying fermentation conditions, such as carbon source, culture time, inoculation volume [11], the culture technique used (static or agitated) [14], and the drying method [15].

Another interesting property of BC is its rehydration ratio, which represents the degree to which removed water (by a dehydration process) is replaced again by water [16]. Dehydration of BC improves its storage and shelf life; nevertheless, it exhibits low rehydration ability after drying [17]. The hydrophilic properties of BC are influenced by many factors such as the drying method, crystallinity in its interior surface area, and the addition of different compounds interfering with its network assembly. Thus, the use of different strategies to modify this property, such as the addition of gelatin, peptides, and soluble polysaccharides to the culture medium during BC biosynthesis, has been studied [18].

Other factors have been evaluated in order to modify other BC structural features and properties. In this regard, the effect of exposing *K. xylinus* culture to a magnetic field or UV light, as well as adding NaCl or antibiotics to the medium, was studied by our group [19]. In this research, results clearly indicated that these conditions promoted the enhancement of the BC crystallinity and thus could also have an effect on other properties such as specific area, porosity, and rehydration rate, but it has not been investigated so far. Likewise, those stimuli might affect BC biosynthesis (yield and production rate), since they are considered cell stressors and thus could trigger BC production as a protection response; however, as far as we know this has not been further studied either. As a result, the evaluation of these changes becomes highly relevant to understand how to obtain BC with the desired characteristics or properties since it might trigger a wider range of applications. In addition, it is also desirable that the stimuli applied do not decrease its production.

Thereby, in the present study, the main objectives were to evaluate the modification in vivo of the BC microporous structure and the fermentation process under such physical and chemical conditions. For this purpose, BC was biosynthesized by *K. xylinus* in static culture under the effect of different stimuli (magnetic field, UV, NaCl, and chloramphenicol), previously studied by our research group [19] but now focused in their effect on the polymer yield, production rate, pore volume, surface area and pore diameter. These changes in the BC were inspected by N_2_ adsorption (BET), scanning electron microscopy (SEM), and infrared spectroscopy (FTIR) measurements. In addition, the effect of the microporous structure vs. crystallinity in BC samples was tested by the rehydration ratio method.

## 2. Materials and Methods

### 2.1. Microorganism and Culture Medium

*A Komagataeibacter xylinus* (strain 2004) was purchased from DSMZ-German Collection of Microorganism and Cell Cultures GmbH, Leibniz Institute, Germany. The strain was reactivated in liquid Hestrin and Schramm (HS) medium, and then streaked on HS agar [20] (glucose, 20 g/L; yeast extract, 5 g/L; peptone, 5 g/L; agar, 15 g/L; pH 6). Agar plates were incubated at 30 °C for 3 days. Next, single colonies were used to inoculate culture tubes with 1.5 mL of HS liquid medium, which were incubated statically for 2 days at 30 °C. After BC pellicles were produced, the culture tubes were agitated vigorously, then 0.5 mL of sterile glycerol solution (60% *w/v*) was added to each of them and stored at −20 °C.

### 2.2. Inoculum Development

HS medium (50 mL in a 250 mL Erlenmeyer flask) was inoculated with 2 mL of *K. xylinus* (frozen stock) and incubated for 48 h at 30 °C and 150 rpm in a rotatory shaker. Then, the produced BC pellicles were disrupted and dispersed under sterile conditions, with a homogenizer (Ultra-Turrax T10 Basic S1, IKA Works Inc., Wilminton NC 28405, USA) during 1 min at a speed level of 6).

### 2.3. Biosynthesis of BC by K. xylinus under the Effect of Physical and Chemical Stimuli

*K. xylinus* cultures were prepared in 100 mL polypropylene flasks (6 cm diameter) with 22 mL of HS medium and inoculated with 3 mL of the previously mentioned inoculum. Static culture was used at 30 °C. The stimuli were applied in two intensity levels for 12 days (in vivo): exposure to UV light (exposure dose, 244 kJ/m^2^ [21]; wavelength, 254 or 366 nm; exposure period, every two days; exposure time, 30 s; distance from opened culture flasks in a laminar flow hood, 30 cm); exposure to low-intensity magnetic field (1 Tesla magnet discs placed at the bottom of the culture flask, one or two; magnet dimensions, 45 × 5 mm; magnetic field strength applied, 0.13 T); NaCl addition (5 or 20 g/L); and antibiotic addition (chloramphenicol, 0.1 or 0.25 mM). Media samples (1 mL), taken below the BC pellicle, were carefully withdrawn (to avoid BC pellicle sinking) every day and analyzed for glucose consumption and pH changes. After 12 days of culture, BC membranes were withdrawn, purified, dried, and weighed. Cultures and analysis were done in duplicate.

For kinetic studies, the level of each stimulus that promoted the highest BC biosynthesis (previous experiment) was selected, and *K. xylinus* was cultured as abovementioned for 16 days. Each treatment (and the control) consisted of nine flasks (with a replicate), one of each was withdrawn at time 0 and every 2 days for determining pH, quantifying glucose concentration and BC production (dried weight of purified membranes).

The maximum BC production, the polymer production rate, and the substrate consumption rate during the fermentation process were calculated using Kaleida Graph Version 4.0 (Synergy Software, Reading, PA, USA) by a modified Gompertz equation [22]:$$P=P_{max}\;.\exp\;\{-\exp\;[\frac{R_{max}\;e}{P_{max}}\;\left(\lambda -t\right)+1]\}$$
where *P* is product concentration (g/L) or substrate consumption (g/L), *P_max_* is maximum product concentration (g/L) or maximum substrate consumed (g/L), *R_max_* is maximum BC production rate (g/L/d) or maximum substrate consumption rate (g/L/d), λ is the lag-phase time (days), and t is the time of fermentation (days).

### 2.4. Glucose, BC and Bacterial Biomass Quantification

For glucose consumption analysis, culture samples (1 mL withdrawn below the BC pellicle) were centrifuged at 10,000 rpm for 10 min. The supernatant was recovered and used for glucose analysis by DNS assay [23].

For BC quantification, pellicles were withdrawn from culture flasks, rinsed with water, and vortexed vigorously for 1 min (two times). Then, BC pellicles were transferred to glass tubes and purified by adding 30 mL of 0.1 M NaOH and heating at 90 °C for 30 min. Finally, the purified BC pellicles were rinsed with water several times until the pH of rinsing water was 7. BC pellicles were dried at 30 °C in a vacuum oven for 2 days and weighed.

The total bacterial biomass was evaluated at the end of the fermentation and it was taken to be the difference between the weights of the dried unpurified BC and the dried purified BC (after treatment with NaOH) [24].

All the determinations were performed in duplicate.

### 2.5. BC Characterization

Purified and dried BC membranes were characterized as follows:

Surface area and pore volume determination by BET analysis. Nitrogen adsorption-desorption isotherm was measured using an ASAP 2020 KMP (Micromeritics, Norcross, GA, USA) N2 adsorption equipment at 77 K. Before the adsorption experiments, each sample was degassed and dried at 423 K for 6 h under vacuum to ensure a clean, dry surface free of any loosely adsorbed species. The Brunauer–Emmett–Teller (BET) surface area was calculated using adsorption data in the relative pressure range of 0.15–0.26, included in the validity domain of the BET equation. The pore size distribution was determined from the adsorption branches of isotherm using the Barrett–Joyner–Halenda (BJH) method, and the total pore volume was determined by nitrogen adsorption P/P0 = 0.995.

### 2.6. Determination of Rehydration Ratio

For determining the rehydration ratio of BC samples, a duplicated experiment was used in which wet BC membranes (purified) were drained at 25 °C for 2 min and weighted (wBC). Next, the membranes were dried at 50 °C until constant weight (dBC), and then dBC samples were immersed in deionized water for 7 h. Finally, the rehydrated membranes (rBC) were weighted [25]. The BC rehydration ratio was calculated as:Rehydration ratio (%) = ((rBC − dBC)/(wBC − dBC)) × 100

### 2.7. SEM Analysis

BC membranes were coated with gold (using an ion sputter coater), and their morphology was analyzed by scanning electron microscopy (SEM) with Tescan Mira 3 LMU equipment (HV, 10 kV; magnification, 20 kx; working distance, 12.68 nm).

### 2.8. FTIR Analysis

All BC samples were chemically characterized by FTIR spectroscopy in a PerkinElmer Spectrum Two spectrometer equipped with an ATR (attenuated total reflection) device. Spectra were performed at 32 scans within spectral window of 400 to 4000 nm^−1^ under 4 cm^−1^ gap of sensitivity.

## 3. Results and Discussion

### 3.1. Effect of Physical and Chemical Stimuli on BC Biosynthesis by K. xylinus

The amount of each BC synthesized by *K. xylinus* exposed to the different stimuli (at two levels of intensity each) is shown in Table 1. We observed that when growing in regular conditions—control HS medium without stimuli (BC ctrl)—the amount of BC produced was 0.81 g/L, which is in concordance to those values reported in other similar studies with this strain: 0.6 [26], 0.83 [27], and 0.85 [28]. Regarding the biosynthesis of BC under the different stimuli applied, we observed that the yield was enhanced by 1.34-, 2.28-, 1.12-, and 1.7-fold (compared to the control) when exposing *K. xylinus* culture to UV light at 254 and 366 nm (BC + UV), using NaCl at 5 g/L (BC + NaCl), and adding chloramphenicol at 0.25 mM (BC + ANT), respectively. After applying the ANOVA and LSD Fisher tests, the treatments that presented statistically significant results (α = 0.5) for triggering an increment in BC yield were: BC + UV (366 nm) and BC + ANT (0.25 mM). This increase in BC biosynthesis seems to be related to a cell protection mechanism triggered by its exposure to such stressors. In relation to UV light, it generates reactive oxygen species that affect microbial growth by disturbing replication and transcription, as well as causing lipid and protein damage [29]. It has been proposed that cellulose synthesis by bacteria plays an important role in protecting the cell from UV irradiation because of the pellicle’s opacity [30]. In contrast, the biosynthesis of BC under a static magnetic field (BC + MF) was unaffected compared to the control (*p* > 0.5). Different magnetic field types and intensities may promote, inhibit or fail to produce any effect on cell metabolism. The positive effect might occur because the magnetic field increases the cell membrane permeability [31] (enhancing nutrients uptake), promotes ATP synthesis, and stimulates enzyme activity [32]. Quan et al. [33] used static magnetic fields at lower intensities of 45 mT, 110 mT and 140 mT on *K. xylinys* cells, finding that the BC production yield was not disturbed under those conditions. On the other hand, Fijałkowski et al. [34] concluded that the exposure to a rotating magnetic field, instead of static, could increase the production yield of BC by *K. xylinus*.

As for chloramphenicol, this antibiotic penetrates the bacterial cells and binds to ribosomes, inhibiting protein synthesis and thus affecting cell growth [35]. Therefore, its presence in the culture medium in a low concentration might have stimulated the biosynthesis of BC as a protective barrier against this compound, as reported for other exopolysaccharides [36].

On the other hand, growth and BC production were significantly (*p* < 0.5) inhibited in the presence of 20 g/L of NaCl, and bacterial growth was strongly affected by such salt concentration, since biomass production was barely noticed under that condition (Table 1). It is known that some bacteria are sensitive to salts such as NaCl because it exhibits specific ionic and water binding properties [37], creating an extracellular hyperosmotic pressure, inducing efflux of the water within the cell that increases the intercellular ionic strength, inhibiting enzymatic reactions [38]. In addition, it has been reported that when the quantity of salt is increased up to 5 g/L, some bacteria are no longer able to consume the available carbon and nitrogen sources in an efficient way, and biomass production is less important. Furthermore, cells appeared to use the available energy for maintenance rather than for cell growth and multiplication.

Therefore, depending on the stimuli and the concentration or intensity level applied, production of BC was either enhanced or remained unaffected (compared to control), which is one of the important aspects to consider since the main objective of this study was to modify BC structure in vivo without affecting its production efficiency.

As for other culture features, the pH decreased in all treatments (3.5–5.1) and the glucose consumption was an average of 80%, except for the culture exposed to NaCl at 20 g/L, which presented the lowest consumption (23.64%) related to the growth inhibition formerly mentioned.

The kinetic profiles of BC production (treatments with higher productivity) are depicted in Figure 1. It was observed that in the treatments with chloramphenicol (0.25 mM) and magnetic field (1 T magnet), as well as in the control, 80% of the total amount of BC synthesis (Figure 1a) occurred during the first 6 days of culture, which agrees with the glucose consumption (Figure 1b) and pH profiles (Figure 1c). This behavior was previously noted when the BC production is high at the beginning of the fermentation process and slows down after 1 week or 10 days. As depicted in Figure 1b, it was also observed that the rate of glucose consumption decreased after 6 days of culture. This might be related to bacterial growth inhibition due to pH (less than 4), affecting BC biosynthesis as well. It is known that pH could greatly influence the biochemical activities of microorganisms [39] when they are exposed to levels beyond their optimum range. Here, the need for maintenance energy (for pH control) increases, leading to a decrease in microbial growth.

Table 2 depicts the kinetic parameters of *K. xylinus* in static fermentation and BC production yields under the effect of the different stimuli. For this experiment, the concentration or intensity level for each stimulus was selected according to the results in Table 1, considering that such conditions did not affect BC yield (presence of 1 T; addition of NaCl, 5 g/L) or even stimulate it (exposure to UV light at 366 nm; addition of 0.25 mM of chloramphenicol). The cultures were replicated under these specific conditions to follow the kinetic profile of BC production, sugar consumption, and pH changes. We obtained similar response to that of the previous experiment: the maximum production of BC was obtained in the treatments with UV (BC + UV) and chloramphenicol (BC + ANT), which are 2.17- and 1.64-fold higher than BC control (*p* < 0.5). The other treatments did not cause an increment in BC production, but it is significant to emphasize that did not affect it. This is in agreement with the main objective of this research, which was to modify the characteristics of BC without decreasing the yield (based on the BC control). As was explained earlier, this increase in BC biosynthesis might be related to the bacteria protection mechanism against UV radiation and harsh chemical environments [40].

The yield coefficient (Y_P/S_) of BC produced vs. substrate consumed is similar to that obtained in other studies [17,41]. It is worthy of notice that the exposure of *K. xylinus* culture to UV light (366 nm) and the addition of chloramphenicol (0.25 mM), did not affect the yield of BC production, but rather improve it. These results suggest further studies to evaluate the application of these conditions as a strategy to increase the BC production in large-scale fermentation.

Regarding the glucose consumption, it was observed that when cultivated under the effect of the magnetic field, NaCl, or chloramphenicol, *K. xylinus* presented a higher consumption rate compared to the control. This increase in the utilization of glucose under stress factors, such as the presence of NaCl (osmotic stress), has been reported previously for other bacteria [42]. To overcome osmotic stress, cells require energy or carbon consumption [43]. Thus, this requirement for additional energy and carbon could explain the higher glucose consumption rates at low salt concentrations. This behavior under stress conditions could also explain the high glucose consumption rate by *K. xylinus* when growing in the presence of chloramphenicol. Concerning the increase of *K. xylinus* glucose consumption under the effect of a magnetic field, it has been mentioned that this factor has a positive effect on bacterial growth [44]. The cause of this result can be explained by an increase of glucose entering through the cell membrane as a consequence of the stimulated transport system, as well as by shortening of the lag phase and excitement of the log phase. It has also been found that, although the glucose consumption rate increases under the effect of magnetic fields, cellular growth might not present a proportional increase because a portion of energy (ATP) (originating from that substrate consumption) will be destined to the maintenance of the bacteria vital functions [45,46].

In the case of the BC production rate, the value obtained for the control is similar to that reported in other studies in similar conditions [47]. On the contrary, all the cultures subjected to the stimuli presented a decrease in the rate of biosynthesis (*p* < 0.5), especially in the treatments BC + NaCl, BC + ANT, and BC + UV (60, 41, and 39% in comparison to BC control). This effect might be related to a decrease in metabolism under harsh environmental conditions (as observed in other bacteria) [48], affecting the ATP production [8].

### 3.2. Effect of Physical and Chemical Stimuli on BC Properties

#### 3.2.1. Specific Area and Pore Volume of BC

The results of the BET analysis of BC are shown in Figure 2. The N_2_ adsorption–desorption isotherms show that all curves of BC samples were type IVa, attributed to mesoporous materials [49]. Characteristic features of this type of isotherm are its hysteresis loop, which starts to occur when the pore size exceeds a critical width wider than 4 nm [49,50]. This hysteresis presents a final saturation plateau, which is related to capillary condensation taking place in mesopores. Finally, the hysteresis loop of all materials is Type H2(b), which is associated with pore-blocking [49,50]. Figure 2 also shows the inset graph of the average pore size distribution for each sample. A narrow pore size distribution in the mesopore range can be observed in all materials.

When performing an analysis of each of the isotherms, clear differences can be observed in the relative pressures at which adsorption occurs and the zone in which hysteresis appears for each of the BC samples. For the BC + NaCl material, the hysteresis starts from the relative pressure P/P_o_ > 0.55. On the other hand, BC + UV and BC control (ctrl) materials present their hysteresis at relative pressures P/P_o_ > 0.45, and finally, for BC + ANT and BC + MF samples, hysteresis begins at the relative pressures P/P_o_ > 0.70. This result indicates that the filling of the mesopores occurs mainly at high pressures for all materials in the range of 0.3 < P/P_o_ < 0.9. For pressures greater than this interval, the filling of the macropores begins.

Regarding the specific surface area, the results are presented in Table 3. As found in the literature, the specific surface area of BC films is variable, typically ranging from 20 to 200 m^2^ g^−1^ [51]. The results of the BET analysis obtained in the present work are within the range of those mentioned by other authors [52], who reported an average pore diameter and surface area of dried BC films of 22.4 and 12.62 m^2^ g^−1^, respectively. It is worthy of notice that the drying method of BC membranes also has an effect on its structure: the use of hot air-drying leads to a dense network, in contrast with the freeze-drying method, which causes a more porous structure. The drying method used in the present research was vacuum drying, and the membranes’ surface characteristics obtained were similar to those mentioned by Zhang et al. [15] (cavities and cracks) (Figure 3). Nevertheless, significant differences were observed between each of the oven-dried BC samples (Table 3) obtained under the different culture conditions. The material with the highest specific area and pore volume BC + NaCl (15.72 m^2^ g^−1^ and 0.11 cm^3^ g^−1^, respectively) is 6- and 4.58-fold higher, compared to the one with the lowest values BC + MF (2.55 m^2^ g^−1^ and 0.024 cm^3^ g^−1^, respectively). Those contrasting values are reflected in the SEM analysis (Figure 3), where it can be observed that the sample BC + NaCl has a higher porosity as well as deeper pores (related to its higher specific area and pore volume). Meanwhile, the sample BC + MF presents noticeably fewer pores, which can be associated to less bacteria mobility. Sano et al. [53] proposed that it was possible to control bacterial motion by electric fields during the production of cellulose. Park et al. [54] observed that cell motion of *K. xylinus* (refered as *A. xylinus*) was restricted by an external magnetic field causing the folding of cellulose chains rather than elongation of cellulose fibers. SEM images of the BC produced under such conditions in which bacterial movement is restricted are similar to those observed in the BC + MF (fewer pores). In addition, Quan et al. [33] mentioned that cellulose is a diamagnetic anisotropic material and that a magnetic field is believed to induce fibers’ alignment, preventing the BC and bacteria regular motion. This is likely related to the formation of dense sheets in those treatments with reduced BET surface area [51].

Huang et al. [55], investigating BC production in the *K. xylinus* (strain CGMCC 2955), found that the polysaccharide structure is related to the rate of its biosynthesis: BC porosity decreased as the biosynthesis rate increased (associated to the *galU* gene expression levels). Similar behavior was observed in our study, especially in BC + NaCl and BC + MF samples: at a lower production rate (0.09 g/L/d), the BET value is high (15.72 m^2^ g^−1^), while at a higher rate (0.16 g/L/d) the BET value is low (2.55 m^2^ g^−1^).

#### 3.2.2. Rehydration Ratio and Crystallinity in BC Samples

The methodology and results for the crystallinity of BC samples was previously reported by our group [19], and these data are shown in Table 3, as is the rehydration ratio (RR). It is observed thar RR of the BC produced in the control treatment (BC ctr) was 7.9%, which is within the range reported for this strain under similar conditions: 5.86 [56], 17% [25]. These low values are common since it is known that BC shows poor rehydration after drying [23], and when relating RR to BET area in these BC samples, two treatments are especially contrasting (BC + NaCl and BC + MF). It is observed that a less porous BC (BC + MF) presents a higher RR value than a more porous sample (BC + NaCl): 22.43% vs. 15.75% (see Figure 3 and Table 3). In this regard, numerous factors can affect the hydrophilic properties of BC. Gomes et al. [57] mentioned that porosity and surface area are related to an increment or reduction of empty spaces among the BC fibrils, and thus, more or less water could enter and be adsorbed onto the polymer. Nevertheless, in the present work, the rehydration ratio of BC might be related to its crystalline structure, since the BC + NaCl sample exhibits higher crystallinity than BC + MF, 86.1 vs 81.5. This finding is in accordance with what Huang et al. observed [58]: BC samples that exhibited higher rehydration presented lower crystallinity and vice versa, which means that an increment in the amorphous region of dried BC creates more spaces to hold water molecules and thus facilitates the rehydration process. On the other hand, it is also clear that the water absorption of the BC samples may not be associated only with the degree of crystallinity, as demonstrated with the BC + UV and BC + ANT samples that have similar crystallinity values of 83% (see Table 3) but different rehydration ratio values (7.83 and 17.95%, respectively). These contrasting results showed the complexity of a multifactorial interpretation that must be studied independently and associated to each stimulus used in BC biosynthesis. Furthermore, each stimulus can promote differences, not only in crystallinity but also in fiber assemblies, fiber length, BC production, crystal size, porosity, mechanical properties, and a larger network of hydrogen bonds [19,59,60], making comparison within each other difficult.

In conclusion, it is possible to modulate the rehydration ability of BC by understanding how each stimulus alters the BC network structure during fermentation, which might have an impact on its further application. For example, the water content in BC is important for biomedical applications such as a dressing material, since the moisture content has an effect on the penetration of active substances into the wound and further healing [25].

#### 3.2.3. FTIR Analysis

Figure 4 shows the FTIR spectra of the BC samples produced under different chemical and physical culture conditions. All spectra (Figure 4a) depicted the classical cellulose signals pattern concerning OH stretching vibration of the hydroxyl groups (3400–3200 cm^−1^), CH vibrations of the aliphatic CH and CH_2_ components (2950–2830 cm^−1^), and the C–O stretching vibration (1200–920 cm^−1^) attributed to the alcohol and ether functional groups of the glucopyranose units. Furthermore, small-signal changes due to the treatments are better observed in the expanded FTIR region (1800–1200 cm^−1^) of Figure 4b. Here, the bacterial cellulose spectrum without treatment (BC ctrl) showed signals at 1640 cm^−1^ attributed to absorbed water or even to amide carbonyls from residual proteins [61]. Aliphatic CH bending vibrations from 1455 to 1280 cm^−1^, including OH bending vibrations associated with the signals at 1335 and 1314 cm^−1^, were observed. The last signal at 1207 cm^−1^ is also due to C-H bending of the aliphatic structure [62,63].

Despite the fact that the BC samples produced under different treatments (MF, NaCl, UV, ANT) showed similar profile signals in this region compared with the control, better signal definition concerning shape and intensity could be observed after treatments (i.e., signals between 1427 and 1280 cm^−1^). This can be interpreted as all treatments contributing to an increase in the structural organization in bacterial cellulose, resulting in an improved crystallinity. In fact, this argument is supported by our previous research that reported using XRD and 13C solid-state NMR analysis [19].

Finally, the data achieved by the BC production under physical and chemical stimuli tested onto *K. xylinus* cultures clearly demonstrated the possibility of promoting changes in the cellulose production, on membrane surface, and in the structural organization. Furthermore, the preset work produced evidence that the correct choice and manipulation of certain culture conditions can enhance specific BC properties for specific applications. On the other hand, the deep understanding of every change observed in these BC samples, and how they can be related to the bacteria performance or adaptation under each stimulus, seems to be a big challenge that is out of the scope of the present research. Nevertheless, those findings motivate us to improve our knowledge, searching the small pieces that compose this big puzzle.

## 4. Conclusions

The tested physical and chemical stimuli modified the BC microporous structure in vivo without negatively affecting its yield, and even more, stimuli such as the addition of chloramphenicol and exposure to UV light enhanced 1.7- and 2.28-fold, respectively, its production under the standard culture condition used. It will be important to evaluate those stimuli in a wider range of concentration or intensity and on diverse BC-producing strains. Results demonstrated that the manipulation of culture conditions affected the membrane pore volume, pore diameter specific area and crystallinity. In contrast, RR ability in BC samples could not be directly associated to their crystalline level, surface area or the porous structure. This seems to be a more complex issue because the stimulus conditions in each BC culture triggered different RR properties that can be associated to diverse structural features of the celluloses obtained under these adverse physical and chemical environments. On the other hand, FTIR spectra of the BC samples under stimuli showed better-defined signals between 1800 and 1200 cm^−1^ compared to the BC control, which can be interpreted as an improvement in the structural arrangement of the polysaccharide. This point is also supported with the improved crystallinity values of the stimulated BC samples. Although the controlling mechanism is not yet understood and is likely to be multifactorial and complex, important data arose regarding the effect of these stimuli onto BC properties and how they can be promoted. Thus, more studies are needed to fully elucidate the biochemical and physiological basis of how some culture conditions affect the bacterial growth and BC production kinetic as well as how such parameters relate to BC fiber assembly and membrane network formation, resulting in changes in many physical and even mechanical properties.

## Figures and Tables

**Figure 1 polymers-14-04388-f001:**
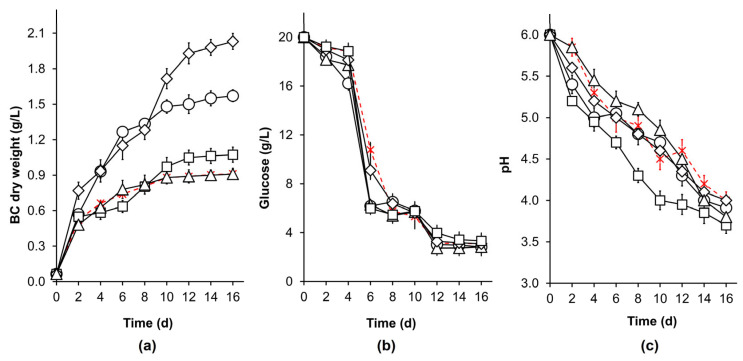
Kinetic of (**a**) BC production, (**b**) glucose consumption and (**c**) pH changes in *K. xylinus* static culture (HS medium). Effect of: UV light, 366 nm (◇); Static magnetic field, 1 T magnet (△); NaCl, 5 g/L (□); and chloramphenicol, 0.25 mM (◯). BC production in the same medium without stimuli (control) is depicted with the dotted line in red color (x).

**Figure 2 polymers-14-04388-f002:**
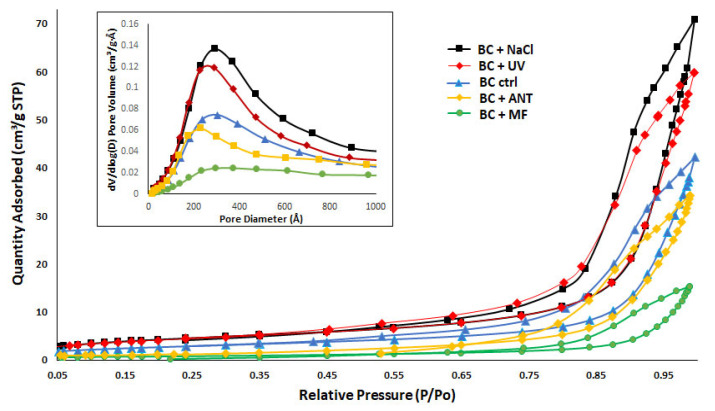
Nitrogen adsorption–desorption isotherms (inset: pore size distribution graph).

**Figure 3 polymers-14-04388-f003:**
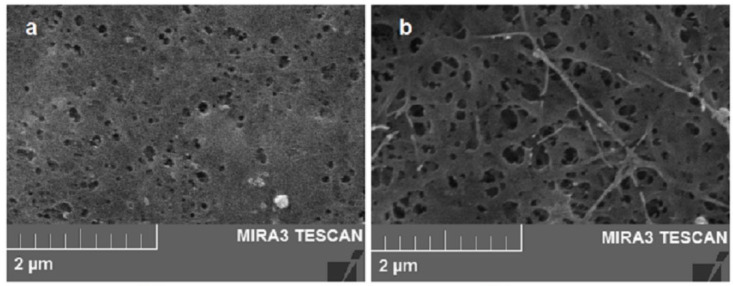
SEM images: (**a**) BC + MF sample with the lowest and (**b**) BC + NaCl sample with the highest specific area and pore volume.

**Figure 4 polymers-14-04388-f004:**
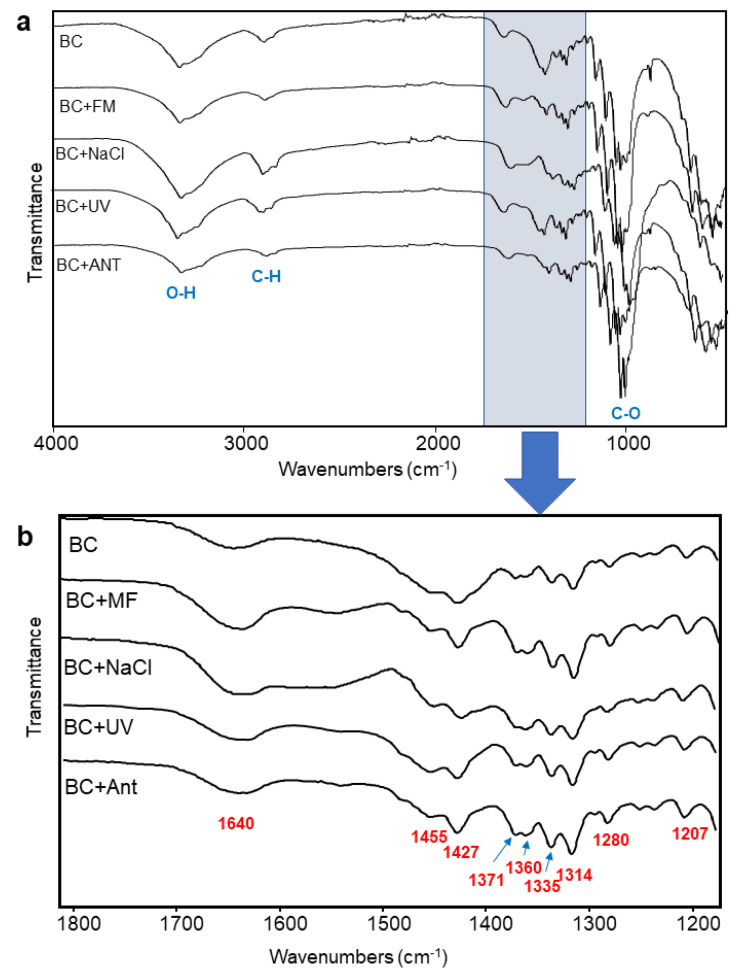
FTIR spectra of (**a**) bacterial cellulose (BC) with and without treatment and (**b**) its expanded region from 1800 to 1200 cm^−1^.

**Table 1 polymers-14-04388-t001:** BC produced by *K. xylinus* exposed to physical and chemical stimuli for 12 days in static culture.

Chemical/Physical Stimuli	Exposition Level	BC (g/L) ^a^	Bacterial Biomass (g/L) ^a^	Glucose Consumption (%) ^b^	Final pH ^c^
Control		0.81 + 0.12	2.76 + 0.3	84.35 + 2.14	4 + 0.1
UV light (nm)	254	1.09 + 0.28	2.72 + 0.4	88.77 + 1.35	3.5 + 0.2
	366	1.85 + 0.13	3.01 + 0.5	84.01 + 4.24	4.1 + 0.4
Static magnetic field ^c^	1	0.81 + 0.18	2.79 + 0.4	83.23 + 3.11	3.8 + 0.3
	2	0.79 + 0.11	3.37 + 0.4	79.89 + 2.77	4.1 + 0.4
NaCl (g/L)	5	0.95 + 0.12	0.0 + 0.0	85.56 + 0.33	3.7 + 0.3
	20	0.05 + 0.0	2.31 + 0.2	23.64 + 0.63	5.1 + 0.2
Chloramphenicol (mM)	0.10	0.87 + 0.11	2.34 + 0.5	80.61 + 0.92	4.1 + 0.1
	0.25	1.39 + 0.23	2.48 + 0.4	85.73 + 2.91	3.9 + 0.2

^a^ Dry cell weight; ^b^ Initial concentration, 20 g/L; ^c^ 1 T magnet piece.

**Table 2 polymers-14-04388-t002:** Kinetic parameters for BC production by *K. xylinus* exposed to physical and chemical stimuli in static culture (HS medium).

Physical/Chemical Stimuli	BC (g/L)	BC Production Rate (g/L/d)	Sugar Consumption Rate (g/L/d)	Y_P/S (mg/g)_
BC	0.90 ± 0.04	0.234 ± 0.01	4.1 ± 0.16	56.71 ± 1.89
BC + UV (366 nm)	1.96 ± 0.10	0.141 ± 0.01	4.1 ± 0.20	124.11 ± 2.46
BC + MF (1 T magnet)	0.91 ± 0.01	0.162 ± 0.00	8.9 ± 0.06	58.14 ± 0.56
BC + NaCl (5 g/L)	1.06 ± 0.11	0.092 ± 0.01	6.7 ± 0.53	65.79 ± 4.18
BC + ANT (0.25 mM) ^a^	1.48 ± 0.13	0.138 ± 0.01	5.4 ± 0.47	91.64 ± 7.20

^a^ Chloramphenicol.

**Table 3 polymers-14-04388-t003:** BET analysis, rehydration ratio, and crystallinity of the BC synthesized under the different physical and chemical stimuli.

Sample	BET Surface Area (m^2^ g^−1^)	Average Pore Diameter (nm)	Pore Volume (cm^3^ g^−1^)	Rehydration Ratio (%)	Crystallinity (%) ^a^
BC ctrl	9.81	23.6	0.066	7.85	76.5
BC + UV (366 nm)	15.05	21.37	0.093	7.83	83.2
BC + MF (1 T magnet)	2.55	25.73	0.024	22.43	81.5
BC + NaCl (5 g/L)	15.72	25.44	0.11	15.75	86.1
BC + ANT (0.25 mM) ^b^	4.08	21.42	0.053	17.95	83.1

^a^ Reported previously by our group [19]; ^b^ Chloramphenicol.

## Data Availability

The data presented in this study are available on request from the corresponding author.

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
