# Peer review of "In Vivo Modification of Microporous Structure in Bacterial Cellulose by Exposing Komagataeibacter xylinus Culture to Physical and Chemical Stimuli"

_polymers, 2022, doi:10.3390/polym14204388_

Round 1
Reviewer 1 Report
Review: Polymers-1958759
Title: In vivo modification of microporous structure in bacterial cellulose by exposing Komagateibacter xylinus culture to physical and chemical stimuli
Authors: Yolanda González-García , Juan Carlos Meza-Contreras , José Antonio Gutiérrez-Ortega , Ricardo Manríquez-González *
Synopsis; The authors present data that demonstrates changes to the bacterial cellulose pedicle generated by the bacterium K. xylinus when cultured under different stressors. This experiments are well designed and the data looks good. The findings in the paper are important for the field and provide insight on the role that these cellulose hydrogel biofilms play in the life history of these K. xylinus. Moreover, this work enables further manipulation and functionalization of the nanocellulose matrixes produced by these bacteria thus providing more possibilities for applications of these materials. The writing was clear, and the figure well presented. I have a few comments and questions I would like the authors to address.
Minor Comments/questions:
Line 87: ‘bought from’ not “was bought to DSMZ”
Lines 106-111 ( as well as in the results sections later): Regarding the exposure to UV light, it would be nice to have the radiant exposure calculated, i.e. how much radiant exposure, J/m2 or something like the spectral exposure to determine the amount of energy input into the system. Likewise for the exposure to the magnetic fields as well, under the conditions the authors provide what is the magnetic field strength on the culture. These are difficult to calculate and will facilitate future work into the effects of these type of stress. In the Results and Discussion section it would nice to include a few sentences beyond what was presented regarding details of how these stressors specifically challenge the cell. Sure, these treatments stress the cell. But what is known in bacterial systems (not necessarily K. xylinus) regarding the mechanisms of these stressors. What does chloramphenicol do to a bacterial cell? How does cells respond to UV radiation? How do magnetic field influence biological systems? A few sentences of a discussion regarding this facilitate new directions for controlling bacterial cellulose production.
Line 163-166: SEM analysis, provide scanning parameters, beam strength working distance, etc.
Line 260 “…. than BC control (p-value<0.5). T><0.5).” the P value probably referred to is P<0.05.
Line 338: Should A. xylinus should be K. xylinus?
Author Response
Manuscript: polymers-1958759
Note: Changes along the corrected version are highlighted in yellow color
Minor Comments/questions:
- Line 87: ‘bought from’ not“was bought to DSMZ”
Answer: It was corrected
- Lines 106-111 ( as well as in the results sections later): Regarding the exposure to UV light, it would be nice to have the radiant exposure calculated, i.e. how much radiant exposure, J/m2 or something like the spectral exposure to determine the amount of energy input into the system.
Answer: Agree with the reviewer´s suggestion and this information was included (line 109-111)
- Likewise, for the exposure to the magnetic fields as well, under the conditions the authors provide what is the magnetic field strength on the culture. These are difficult to calculate and will facilitate future work into the effects of these type of stress.
Answer: We agree and data was added (lines 111-113)
- In the Results and Discussion section it would nice to include a few sentences beyond what was presented regarding details of how these stressors specifically challenge the cell. Sure, these treatments stress the cell. But what is known in bacterial systems (not necessarily K. xylinus) regarding the mechanisms of these stressors. What does chloramphenicol do to a bacterial cell? How does cells respond to UV radiation? How do magnetic field influence biological systems? A few sentences of a discussion regarding this facilitate new directions for controlling bacterial cellulose production.
Answer: Agree with the reviewer´s concern and the information were included along the manuscript (lines 192-195, 198-203, 207-209, 232-235)
- Line 163-166: SEM analysis, provide scanning parameters, beam strength working distance, etc.
Answer: Yes, the information was provided (line 171)
- Line 260 “…. than BC control (p-value<0.5). T>0.5).” the P value probably referred to is P<0.05.
Answer: Yes, it was corrected (line 276).
- Line 338: Should A. xylinus should be K. xylinus?
Answer: The research performed by Park et al. was in 2012, at that time the strain was known as A. xylinus. The manuscript was modified to explain this issue (line 359-360).

Reviewer 2 Report
The article "In vivo modification of microporous structure in bacterial cellulose by exposing Komagateibacter xylinus culture to physical and chemical stimuli" presents new data on changes in the physicochemical and structural properties of bacterial cellulose (BC) obtained in the presence of substances "NaCl, chloramphenicol" or with exposure to "low-intensity magnetic field and UV light" directly in the process of BC biosynthesis. The main discussed properties of BC are specific area and porosity (“specific area, pore-volume”). The authors substantiate the need to control these properties while maintaining the productivity of BC. In general, the hypothesis is very attractive and can be claimed by many researchers. The article attracts with its honesty: the synthesis of BC with low productivity is described, the consequences of the impact on the biosynthesis of physical and chemical factors. Quite frankly, I think that the strongest side of this article is the conclusion “the controlling mechanism is not yet understood”. The authors conducted a large amount of experimental work for only one strain and showed that the properties do change, albeit slightly, while recording in detail not only the yield and rate of BC formation, but also the yield of the total biomass. Figures and tables in good quality. The article is easy to read, easy to understand, conducive to discussion.
Questions and remarks:
1. The authors must follow the abstract, the text of the article, the order of presentation for physical and chemical factors. If we first describe the physical, then everywhere, including the annotation, materials and methods, the conclusion, the physical must be the first.
2. The influence of crystallinity on the "rehydration ratio" is presented rather unconvincingly, since the crystallinity varies (increases from 76.5 to 86.1%) in a very narrow range, and the rehydration ratio has unexpected values.
3. Conclusions are made more for the future than for the essence of the research.
4. The article lacks an explanation of the reason for changing the specific surface area of ​​the aircraft.
5. After the article has already been submitted, it is possible that the authors have a suggestion: for what reason they selected these physical effects and these substances.
6. The materials and methods describe the preparation of VS samples for analysis; there is no freeze-drying. Is this really true? Are the BC samples dried under vacuum? But then they should have "collapsed" more strongly.
Author Response
Manuscript: polymers-1958759
Note: Changes along the corrected version are highlighted in green color
Questions and remarks:
- The authors must follow the abstract, the text of the article, the order of presentation for physical and chemical factors. If we first describe the physical, then everywhere, including the annotation, materials and methods, the conclusion, the physical must be the first.
Answer: Corrections were done along the manuscript
- The influence of crystallinity on the "rehydration ratio" is presented rather unconvincingly, since the crystallinity varies (increases from 76.5 to 86.1%) in a very narrow range, and the rehydration ratio has unexpected values.
Answer: Agree with reviewer´s comment and more discussion was added in order to explain this issue as well as the intrinsic complexity of each BC obtained (lines 404-416).
- Conclusions are made more for the future than for the essence of the research.
Answer: Agree with the reviewer´s concern and conclusions were modified (lines 456-469).
- The article lacks an explanation of the reason for changing the specific surface area of ​​the cellulose
Answer: Agree with the reviewer´s comment and reasons are explained in lines 48-53.
- After the article has already been submitted, it is possible that the authors have a suggestion: for what reason they selected these physical effects and these substances.
Answer: We agree with the suggestion and the reasons were included (lines 66-73).
- The materials and methods describe the preparation of VS samples for analysis; there is no freeze-drying. Is this really true? Are the BC samples dried under vacuum? But then they should have "collapsed" more strongly.
Answer: BC samples were not freeze-dried. As you mentioned, the drying method has an influence on the structure and properties of BC, it was briefly mentioned in the manuscript (Line 56). Nevertheless, more information was added to the manuscript to address this issue (lines 61, and 344-351)
